# Targeting SAM-I Riboswitch Using Antisense Oligonucleotide Technology for Inhibiting the Growth of *Staphylococcus aureus* and *Listeria monocytogenes*

**DOI:** 10.3390/antibiotics11111662

**Published:** 2022-11-19

**Authors:** Martina Traykovska, Robert Penchovsky

**Affiliations:** Department of Genetics, Faculty of Biology, Sofia University St. Kliment Ohridski, 8 Dragan Tzankov Blvd., 1164 Sofia, Bulgaria

**Keywords:** gram-positive bacteria, design of antisense oligonucleotides, cell-penetrating peptides, antibacterial agents, drug targets, riboswitch, antibacterial resistance, antibacterial drug discovery

## Abstract

With the discovery of antibiotics, a productive period of antibacterial drug innovation and application in healthcare systems and agriculture resulted in saving millions of lives. Unfortunately, the misusage of antibiotics led to the emergence of many resistant pathogenic strains. Some riboswitches have risen as promising targets for developing antibacterial drugs. Here, we describe the design and applications of the chimeric antisense oligonucleotide (ASO) as a novel antibacterial agent. The pVEC-ASO-1 consists of a cell-penetrating oligopeptide known as pVEC attached to an oligonucleotide part with modifications of the first and the second generations. This combination of modifications enables specific mRNA degradation under multiple turnover conditions via RNase H. The pVEC-ASO targets the S-adenosyl methionine (SAM)-I riboswitch found in the genome of many Gram-positive bacteria. The SAM-I riboswitch controls not only the biosynthesis but also the transport of SAM. We have established an antibiotic dosage of 700 nM (4.5 µg/mL) of pVEC-ASO that inhibits 80% of the growth of *Staphylococcus aureus* and *Listeria monocytogenes*. The pVEC-ASO-1 does not show any toxicity in the human cell line at MIC80’s concentration. We have proven that the SAM-I riboswitch is a suitable target for antibacterial drug development based on ASO. The approach is rational and easily adapted to other bacterial RNA targets.

## 1. Introduction

The decreasing effectiveness of antibiotics in treating infectious diseases has become a global issue that causes millions of human deaths annually [1]. Moreover, the high rates of antibiotic usage in hospitals, communities, and agriculture have contributed to the rise of multidrug-resistant (MDR) bacterial strains. Over the last two decades, the number of pathogenic bacterial strains resistant to one or more antibiotics has substantially increased. For instance, the methicillin-resistant strains of *Staphylococcus (S.) aureus* (MRSA), *Escherichia (E.) coli* ST131, and *Klebsiella* ST258 are estimated to cause many deaths in the USA only. A 2019 year report from a United Nations (UN) interagency group reported that drug-resistant diseases have already caused at least 700,000 deaths globally, with 230,000 deaths from MDR tuberculosis. The figure may increase to 10 million deaths globally annually by 2050, under the worst scenario, if adequate actions are not taken. This projected figure must prompt the antibacterial drug development community to produce new antibacterial drug agents that the pharmaceutical industry can turn into novel antibiotics.

Bacteria have developed various mechanisms for overcoming the action of the antibiotics, such as efflux of the antibiotic molecule via transport proteins, degradation or enzymatic modification of the antibiotics, and receptor mutations. Eventually, bacteria can develop antibiotic resistance (AR) to any widely used antibiotic. Therefore, we must develop novel strategies for developing novel antibiotics that must be adaptive to tackle any new AR that may arise in the future [2]. We must employ new molecular targets, technologies, methods, and strategies to achieve this goal [3].

Our research in this paper is focused on developing new efficient strategies for antibacterial drug discovery that may lead to a shorter development time and a higher success rate of drug candidate discovery. Our approach combines synthetic biology methods, bioinformatics [4,5], and rational drug design of antisense oligonucleotides (ASOs) [6,7,8,9,10,11,12,13,14]. Herein, we employ ASO that binds explicitly with the complementary sequence of the S-adenosyl methionine (SAM-I) riboswitch located in the 5′-untranslated region (UTR) of mRNAs and inhibits the growth of pathogenic human bacteria, such as *S. aureus* and *Listeria (L.) monocytogenes* [15].

Riboswitches are gene control units found in the genomes of many bacteria but absent in humans, which makes some of them suitable targets for antibacterial drug development [16,17]. They have two parts: an aptamer domain and an expression platform. The aptamer domain selectively binds specific metabolites or ions that undergo conformational changes, leading to a change in the expression platform that alters the gene expression, mainly by transcriptional termination or prevention of translation. Until now, over 40 riboswitch classes have been found to regulate gene expression in bacteria and still counting [18]. The SAM-I riboswitch is found in nine Gram-positive human pathogenic bacteria (Appendix A). It is a highly conserved RNA located in the 5′-UTR of mRNAs coding proteins involved in the biosynthesis of methionine or cysteine in bacteria [19]. This riboswitch is also known as S-box or S-box leader. SAM is an essential coenzyme not only for bacteria but also for all other organisms. It is synthesized directly from methionine by SAM synthetase and serves as a donor of methyl groups for modifications of proteins and nucleic acids. The main synthetic pathway for methionine and cysteine production are evolutionary conservative in many bacteria.

Several SAM riboswitch types are classified by their aptamer domains, including SAM-I, SAM-II, SAM-III, and SAM-IV riboswitches. They all specifically bind SAM or its derivate and control sulfur metabolism in many bacterial species. The SAM-I riboswitch comprises four branches connected to two sets of coaxial helixes arranged next to each other. They are connected through a loop at the end of stem P2 and J3/4 junctions. SAM binds with the pocket formed by helix P1, P2, and J1/2 [20,21]. The ligand acquires compact conformation by binding the bulge of the P3 via hydrogen bonds and conservative guanine in J1/2, which stabilizes by the binding of methionine with adenine [21]. The sulfur recognizes the methyl group, which forms an electrostatic interaction with negative surface potential created by tandem AU couples in the minor groove of P1. These pairs are highly conservative, and changes in their rotation and identity (GC pairs instead of AU) result in reducing the binding affinity of SAM [19,22].

Most bacteria and plants synthesize the amino acid methionine de novo after the initial steps of assimilation of inorganic sulfate and synthesizing cysteine or homocysteine. L-methionine plays an essential role in cellular metabolism, particularly in the biosynthesis of peptides. The synthesis of all polypeptide chains starts with including methionine at their N-terminus. The derivate of methionine, SAM, serves as a donor of methyl groups, hence the vital role of methionine in cellular methylation processes. The biochemical pathway of synthesis of SAM includes genes that are part of the glycine, serine, and tyrosine metabolism and from the aspartate metabolism producing methionine (Appendix A).

Therefore, blocking the synthesis of SAM should inhibit the division of bacterial cells. In bacteria, the SAM-I riboswitch controls gene expression through transcriptional termination (Appendix A). Therefore, the SAM-I riboswitch is a suitable target for antibacterial drug discovery since it controls biosynthetic pathways for SAM, which cannot be avoided.

## 2. Results

### 2.1. Design of Antisense Oligonucleotide Targeting SAM-I Riboswitches

We designed our ASO to be complementary to the aptamer sequence of the SAM-I riboswitch, which is part of the S-box operon found in the genomes of *S. aureus* and *L. monocytogenes* (Figure 1A). We used a combination of first- and second-generation modifications to design our ASO that targets the SAM-I aptamer domain in S-box mRNAs.

In the first generation of ASO modifications, the oxygen atom linked to the phosphodiester bond phosphate is replaced by the sulfur atom to make so-called PS-modification, which allows the activation of RNase H. In the second ASO generation, the H atom at the 2′-position of the ribose is replaced by the -O-CH_3_ group, which increases the stability against endo-and-exo-nucleases more than PS-modification. However, the second generation of ASO modification inhibits the RNase H cleavage. Therefore, we applied the second-generation ASO modification at the ends, while the PS modifications were used in the middle of the ASO (Figure 1). This enabled the cleavage of the S-box mRNA by the RNase H endonuclease under multiple turnover conditions.

The chimeric ASO-1 is conjugated with pVEC. The pVEC can pass through the cell membrane in prokaryotes and eukaryotes. It is not toxic by itself in all concentrations used. It is composed of 18 amino acid residues (AAR) [23]. The N-terminus is hydrophobic, the middle part is positively charged as it is rich in Arg, and the C-terminus is hydrophilic. pVEC peptide is derived from murine vascular endothelial-cadherin protein (VE-cadherin), whose primary function is to make physical contact between the neighboring cells. VE-cadherin plays a role in the control of vascular permeability and angiogenesis. pVEC is not toxic for the cells and is degraded by peptidases. 

After the pVEC-ASO-1 enters the bacterial cell, only the oligonucleotide part of pVEC-ASO-1 hybridizes to its complementary sequence of the aptamer domain of the SAM-I riboswitch (Figure 1B). The formed double-stranded RNA/DNA hybrid triggers enzymatic digestion by the endonuclease RNase H (Figure 1C). The pVEC is needed to deliver the chimeric ASO-1 into the cell, after which the intracellular peptidases degrade it. That does not impact the ability of the ASO-1 to trigger the specific degradation of S-box mRNA. The S-box mRNA cleavage leads to a lack of translation of the S-box operon, as mRNA is subject to enzymatic digestion by RNase H (Figure 1D).

The design of the pVEC-ASO-1 is based on several criteria to guarantee the efficient inhibition of the target RNA. They include the targeted region of the RNA sequence to be single-stranded, ASO-1 not to have significant similarities with the expressed human RNAs, not to form stable secondary structures (Appendix A), and not to form stable dsRNAs with themselves. Thus, ASO-1 is to be accessible for hybridization with the target RNA domain above the thermodynamic ensemble of 10 kcal/mol according to the partition function of the RNAfold program of the Vienna RNA folding server. For the reasons mentioned above, the ASO does not create stable double-stranded hybrids beyond the thermodynamic ensemble of −10 kcal/mol.

According to our thermodynamic-based calculation (https://penchovsky.atwebpages.com/applications.php?page=47) (accessed on 17 November 2022) [24], the melting point of our ASO/RNA hybrid is 55.3 °C to ensure stability. 

### 2.2. Inhibition of the Bacterial Growth Via the S-box mRNA with ASO That Targets the SAM-I Riboswitches

The inhibitory effect of pVEC-ASO-1, designed to bind to the complementary sequence of the aptamer domain of the SAM-I riboswitch, is demonstrated in cells of unconditional human pathogen *L. monocytogenes* and the conditional human pathogen *S. aureus*. As depicted in the multiple alignment, we chose a conservative region of the aptamer sequence that is common for both pathogens (Figure 2A). In addition, the conditional human pathogen *E. coli* was used as a negative control since the SAM-I riboswitch is not present in this bacterium; therefore, the pVEC-ASO-1 should not inhibit the bacterial growth of *E. coli*. Furthermore, the SAM-I riboswitch-targeted region is single-stranded, making it a particularly suitable target for hybridization with pVEC-ASO-1. Firstly, to prove the inhibitory effect of pVEC-ASO-1 on S-box mRNA, *S. aureus* was incubated till it reached 0.3 OD at 600 nm. Total RNAs were isolated as described in Section 5 using a same master mix for both reactions.

One bacterial cell culture was incubated with 2000 nM pVEC-ASO-1 and the other without pVEC-ASO-1. Next, the isolated RNAs were converted into cDNA by a SMARTScribe reverse transcriptase, as described in the manufacturer’s instructions. Two different PCR reactions were run, as described in the ES using the same master mix for both reactions.

Amplification was observed only on the lane with cDNA incubated without ASO-1 (Figure 2B). In contrast, no amplification was observed in the presence of ASO-1. That demonstrates that ASO-1 inhibited the S-box mRNA. In addition, to ensure that cDNAs were properly synthesized and the PCR amplifications well, we did a positive control of the cDNAs with primers for FMN in S. aureus that produced application of FMN mRNA without or without treatment of pVEC-ASO-1. As expected in both PCR reactions there were amplification (Appendix A).

In the samples containing the highest concentration of pVEC-ASO-1, 2000 nM observed an inhibition of the bacterial growth of *S. aureus* and *L. monocytogenes* measured with a spectrophotometer at 600 nm (Figure 3A). The line with the triangles shows the bacterial growth of *L. monocytogenes* in the absence of pVEC-ASO-1 reached a maximum of about 1.3 OD after an incubation time of 3 h (h) and stayed the same over the next 8 h (Figure 3A). In contrast, for the same bacteria in the presence of 2000 nM ASO-1 (Figure 3A, the line with the upside-down triangles), a maximum of less than 0.3 OD was observed after an incubation time of 4 h and stayed the same in the following 8 h. Without pVEC-ASO-1, the growth of *S. aureus* reached a maximum of approximately 1.3 OD after an incubation time of 4 h (Figure 3A, the line with the rectangles) and stayed the same for the next 8 h. At the same time, the bacterial growth of *S. aureus* in the presence of 2000 nM pVEC-ASO-1 (Figure 3A, the line with the circles) reached a maximum of 0.3 OD after incubation of 4 h and stayed the same for the next 8 h.

At the concentration of 1000 nM, pVEC-ASO-1 inhibited the bacterial growth of *L. monocytogenes* (Figure 3A), while with *S. aureus*, a more potent inhibition was observed (Figure 3B). The line with the upside-down triangles shows the bacterial growth of *L. monocytogenes* in the presence of pVEC-ASO-1, which reached a maximum of less than 0.4 OD after incubation of 4 h and stayed the same in the next 8 h.

The bacterial growth of *S. aureus* and *L. monocytogenes* in the absence of ASO-1 reached a maximum of about 1.3 OD; *L. monocytogenes* reached this maximum after incubation of 3.5 h (Figure 3B, the line with the triangles), while *S. aureus* after 4 h (Figure 3B, the line with rectangles).

At the next concentration of pVEC-ASO-1 of 700 nM, inhibition of the bacterial growth of *L. monocytogenes* and *S. aureus* was observed to be the same as in the previous two concentrations of ASO-1 (Figure 3C). With pVEC-ASO-1, the growth *of L. monocytogenes* reached a maximum bacterial growth of around 0.5 OD after an incubation time of 4 h (Figure 3C, the line with the upside-down triangles), and it did not change in the following 8 h. In the presence of ASO-1, the growth of *S. aureus* (Figure 3C, the line with the circles) reached the maximum of less than 0.4 OD after an incubation time of 3.5 h, and it did not change in the next 8 h. The growth of *L. monocytogenes* (Figure 3C, the lines with the triangles) and *S. aureus* (Figure 3C, the line with the rectangles) in the absence of ASO-1 reached a maximum of 1.3 OD after an incubation time of 4 h. It stayed the same in the following 8 h.

At a concentration of pVEC-ASO-1 of 350 nM, almost no difference was observed in the bacterial growth of *L. monocytogenes* and *S. aureus* in the presence and absence of ASO-1 (Figure 3D). In the presence of ASO-1, *L. monocytogenes* (the line with the upside-down triangles) and *S. aureus* (the line with the circles) reached a maximum of 1.2 OD after an incubation time of 4 h and stayed the same in the following 8 h. The growth of *L. monocytogenes* in the absence of ASO-1 (Figure 3D, the line with the triangles) and *S. aureus* (Figure 3D, the line with the rectangles) is the same as the growth of the bacteria with pVEC-ASO-1.

At the lowest concentration of pVEC-ASO-1 of 150 nM, there was no inhibition of the bacterial growth of *L. monocytogenes* and *S. aureus* (Figure 3E). With pVEC-ASO-1, *L. monocytogenes* (Figure 3E, the line with the upside-down triangles) and *S. aureus* (Figure 3E, the line with the circles) reached the maximum of 1.3 OD after an incubation period of 4 h. They did not change in the following 8 h. Without pVEC-ASO-1, the growth of *L. monocytogenes* (Figure 3E, the line with the triangles) and *S. aureus* (Figure 3E, the line with the rectangles) was the same as with pVEC-ASO-1.

The highest concentration of 2000 nM of pVEC-ASO-1 was also used in bacterial cells of *E. coli* (Figure 3F). Still, bacterial growth was not inhibited as the SAM-I riboswitch is not present in its genome. The kinetics of the bacterial growth of *E. coli* (Figure 3F, the line with the upside-down triangles) is the same as the samples with no pVEC-ASO-1 (Figure 3F, the line with the triangles).

Regarding the inhibitory effect of the tested concentrations, we established MIC80, a standard used to calculate the minimum concentration of the relevant antibiotic molecule for 80% inhibition of bacterial growth based on the curves in Figure 3 after 5 h of incubation with different concentrations of pVEC-ASO-1. For the pVEC-ASO-1 that hybridizes with the complementary sequence of the aptamer domain of the SAM-I riboswitch and controls gene expression of the S-box operon, MIC80 is around 700 nM, 4.5 μg/mL. This concentration includes ASO and pVEC, which can pass through the cell membrane. The pVEC is not toxic for the A549 cell line and is degraded by proteases. Furthermore, pVEC does not have any toxic effect on the bacterial growth of *S. aureus* (Appendix A) [6,7]. These results show that the inhibitory effect of bacterial growth is due only to the ASO, which binds to the complementary sequence of the SAM-I riboswitch. Without pVEC, ASO cannot pass through the bacterial cell membrane and exhibit its inhibitory effect (Appendix A) [6,7]. Only the chimeric molecule consisting of ASO and pVEC can pass through the bacterial cell membrane.

pVEC-ASO-1 at a concentration of MIC80 is 700 nM, 4.5 μg/mL is not toxic for the human cell line of the non-small lung cancer A549 (Figure 4). At a concentration of 2000 nM of the pVEC-ASO-1, the survival of the A549 human cell line is 61%, with only 39% toxicity. At a concentration of 1000 nM of ASO, the survival of the A549 human cell line is 94%, with only 6% toxicity. While the concentration of 700 nM is 99%, no toxicity is observed in the absence of pVEC-ASO-1.

We employed pVEC-ASO-2 with five mismatches compared to pVEC-ASO-1 as a negative control. No inhibition of the growth of *S. aureus* was observed (Figure 5).

## 3. Discussion

Three different generations of ASOs are classified based on the chemical modifications applied [10]. These chemical modifications have improved nuclease resistance, extended the tissue half-life, and reduced non-sequence-specific toxicity. The first generation corresponds to the modification of phosphodiester backbones such as phosphorothioate (PS), where one of the unbound oxygen atoms is replaced with a sulfur atom. These PS-modified ASOs can induce an RNase H-related cleavage of mRNA. The second generation represents oligonucleotides in which the structural modification is not limited to the backbone linkage but includes structural modifications of the ribose: ASO with 2′-O-alkyl modifications of the ribose were developed [11]. These modifications improve binding affinity, increase efficacy, decrease the protein binding of oligonucleotides, and enhance nuclease resistance. The third class of ASO modifications includes locked nucleic acids (LNA) [25] and peptide nucleic acids (PNA). They are far more resistant to enzymatic degradation but also inhibit the action of RNase on their hybrids. 

We chose a combination of the first- and second-generation ASO modifications that enable the pVEC-ASO-1 to have extensive stability against nucleases and to cleave target RNA by activating RNase H under multi-turnover conditions. This combination of ASO modifications and pVEC (Figure 1) as a delivery carrier is well-proven in this paper and three other recently published works. They all target different riboswitches with chimeric pVEC-ASO with very similar MIC80 doses. In addition, we have selected the targeted riboswitches of glmS, FMN, TPP, and SAM-I as a result of vigorous bioinformatics analyses. We target the conservative part of the riboswitch aptamers that should not mutate quickly. However, we know that bacteria can develop resistance to any antibiotics. The most likely scenario in our case is to have mutations in the targeted region of the aptamer, which can quickly be established and fixed by changes in the ASO’s sequences. It is improbable that the bacteria will now invent enzymes capable of hybridizing the modified ASO or becoming impenetrable for CPPs such as pVEC. Our current approach may improve further by employing CPPs that can enter only bacteria.

Small molecules and ASOs can target the aptamer domain of bacterial riboswitches because they have 3D binding pockets for specific metabolites and well-known RNA sequences. Thus, different drug-developing strategies can be applied to the same targets, which can open new avenues of research.

We have used the A549 cell line because *S. aureus* is found in the lung in many infected penitents. Thus, we have applied one of the most used and well-characterized lung cell lines for our ASO toxicity experiments. The A549 cell line was already used for rapid and efficient toxicity testing [26].

## 4. Conclusions

This paper demonstrates the specific inhibition of the SAM-I riboswitch mRNA by pVEC-ASO-1, leading to the growth inhibition of *S. aureus* and *L. monocytogenes*. 

The CPP pVEC attached to the ASO-1 enables its efficient delivery into the bacterial cell. We have established MIC80 at the pVEC-ASO-1 concentration of 700 nM and prove that the pVEC-ASO-1 does not show any toxicity to a human cell line. pVEC-ASO-1 has a bacteriostatic effect in the experimentally tested pathogenic bacteria, *S. aureus* and *L. monocytogenes*, by targeting the synthesis and transport of SAM pathways through the action of RNase H. At the same time, the specific action of pVEC-ASO-1 is proven by the lack of any inhibition in *E. coli*, where SAM-I riboswitch is not present.

This ASO-based riboswitch targeting technology is already proven and published for glmS^6a^ and FMN riboswitches^6b^. Our approach combines bioinformatics analyses [5,11] for the suitability of riboswitches to be antibacterial drug targets with the rational design of chimeric ASOs. We can claim that we have achieved 100% efficiency in inhibiting the growth in vitro of various human pathogenic bacteria by targeting all selected riboswitches.

Designer pVEC-ASOs targeting FMN [7], TPP [8], and SAM-I riboswitches have MIC80 around 700 nM and exhibit a bacteriostatic effect. All designer pVEC-ASOs worked as expected, proving the high fidelity of our rational approach for drug design, including drug-target evaluation [4,5]. FMN, TPP, and SAM-I are essential co-factors for many enzymes in the cell, and it takes 3 to 5 h after stopping their syntheses till the inhibition of bacterial growth. The bacteria may use already synthesized FMN, TPP, or SAM-I within this period of 3 to 5 h. In contrast, the complete inhibition of glmS mRNA by pVEC-ASOs has a bacteriocidal effect due to the inhibition of cellular wall synthesis [6]. By targeting riboswitches with ASOs, we can develop broad [7] and narrow-range [6] antibiotics.

Moreover, our approach possesses some significant inherited adaptation advantages for addressing the future development of AR. The most probable mechanism to avoid the pVEC-ASO-1 action is to mutate in the target region of the SAM-I riboswitch. This can happen even though we target a highly conservative region of SAM-I aptamer^13^. That can be quickly established by DNA sequencing, and the ASO-1 sequence can be altered accordingly. It is improbable that bacteria can develop resistance against ASO-1 by evolving new enzymes for much faster degradation or stopping the pVEC from entering the cell. This adaptation of ASO to a new mutation of the target RNA is much easier to do than with small molecules that target a 3D structure of functional RNA or protein. We can use other chemical modifications or CPP, assuming such improbable events can happen. Thus, we firmly believe that the technology presented may become a valuable new tool in the toolbox of novel methods for antibacterial drug development.

## 5. Experimental Section

### 5.1. Databases for Bioinformatics Analysis

The nucleotide sequences of the SAM-I riboswitch present in the human pathogenic bacteria were taken from the Rfam database 12.0 (http://rfam.xfam.org/) and the RSwitch database (accessed on 17 November 2022) [27]. The pathway of biosynthesis of SAM-I riboswitch was designed using the following databases KEGG PATHWAY and BioCyc. The BioCyc collection of Pathway/Genome Databases (PGDBs) references many sequences. BioCyc also integrates information from bioinformatics databases, such as UniProt protein features and gene ontology information. Basic Local Alignment Search Tool (BLAST) search algorithm (http://blast.ncbi.nlm.nih.gov/Blast.cgi (accessed on 17 November 2022)) was used to detect regions of local similarity between the sequences of the bacteria containing the SAM-I riboswitch. ClustalX (2.0) was used to make multiple sequence alignments and analyze the obtained results. Vienna RNA folding server at (http://rna.tbi.univie.ac.at (accessed on 17 November 2022)) was used to predict secondary structure.

### 5.2. Antisense Oligonucleotide Sequence

The designed ASO-1 represents a chimeric molecule as its 5′-terminus is attached to the carboxyl terminus of the CPP pVEC (LLIILRRRIRKQAHAHSK). The overall sequence of the designed pVEC-ASO-1 is LLIILRRRIRKQAHAKSK-T1C1C1C2T2C2C2A2C2C2A2C1T1C1, where the index “_1_” means to 2′-alkyl modifications of the ribose. In contrast, the index “_2_” means the first generation of modifications with the phosphor-thiolate group (PS) of the phosphodiester bond. These modifications are necessary for recognizing the ASO-1 by the RNase H endonuclease. pVEC-ASO-1 was designed to be complementary to the sequence of the SAM-I aptamer domain and purchased from GeneLink, Inc., Elmsford, NY. We also used pVEC-ASO-2 is LLIILRRRIRKQAHAKSK-T1C1C1C2G2C2A2C2C2A2C2C1T1C1 with 5 mismatches compared to pVEC-ASO-1 and used by us as a negative control.

### 5.3. PCR Amplification

Total RNAs were isolated from *S. aureus* using the QIAprep Spin Miniprep kit (QIAGEN). The total RNAs were isolated from the same amount of cells with 0.15 OD treated with or without pVEC-ASO-1. The isolated RNAs were converted into cDNA using the SMARTScribe reverse transcriptase purchased from Takara following the manufacturer’s manual using a master mix of all chemicals. We ran the PCR reaction with the obtained cDNA containing the following reagents: 10x Taq buffer, 10 mM dNTPs mixture, 0.5µM forward primer-1 5′ATCCTGAGTGGTGGAGGGAC, and reverse primer-2 5′ACGGTTTGGCACCTTTCTTT for the SAM-I riboswitch, 25 mM MgCl_2_, 100 ng DNA, 1,25 µL Taq polymerase enzyme, and sterilized distilled water. The primers were purchased from Sigma-Aldrich/Merck, Darmstadt, Germany, each with a length of 20 nucleotides, while the other PCR reagents were purchased from Thermo Scientific, Waltham, MA. There were master mixes used for all chemicals except for the cDNA in both reactions.

The PCR conditions under which the reaction was run with lead heating at 112 °C, 30 cycles, denaturation at 95 °C for 60 sec., annealing at 55 °C for 90 sec., extension at 72 °C for 60 sec (LBK, Quantarus Q Cycler Gradient PCR, Quanta Biotech, UK). Next, the PCR samples were precipitated with 3.3 M sodium acetate and three volumes of absolute cooled ethanol. They were kept at −80 °C for 10 min and centrifuged at 12,000× *g*. After removing the supernatant, the amplified product was washed with 70% ethanol and dissolved in 10µL sterile water. Next, the samples were loaded in 2% agarose gel. A GeneRuler 100 bp Plus DNA Ladder (Thermo Scientific, Waltham, MA, USA) was used.

The cDNAs were tested as positive controls with PCR with primer-3 5′AAATTCATCTTCGGGGTCG and primer-4 5′CATACTTATTGTGATAAGACGTT. PCR conditions were as follows: lead heating at 112 °C, 31 cycles, denaturation at 95 °C for 60 s, annealing at 55 °C for 90 s, extension at 72 °C for 60 s (LBK, Quantarus Q CyclerGradient PCR, Quanta Biotech, UK) [7].

### 5.4. Bacterial Strains and Media

We used the following bacterial strains: *S. aureus* strain ATCC 25923, *E. coli* strain K12, and *L. monocytogenes* strain ATCC 8932, purchased from DSMZ GmbH (German Collection of Microorganisms and Cell Cultures). Bacteria were cultivated in a Luria-Bertani (LB) medium containing 10 g Bacto-tryptone, 5 g yeast extract, and 10 g NaCl with a pH of 7.5 per liter. Bacteria were incubated overnight at 37 °C until reaching an Optical Density (OD) of 0.8 at 600 nm. They have diluted 200 times again in LB medium in a Memmert incubator for 12 h with and without ASO-1 in five different concentrations: 2000 nM, 1000 nM, 700 nM, 350 nM, and 150 nM. For each concentration, three repetitions were made. The OD was measured every half an hour using a spectrophotometer, Pharmacia Biotech model Ultrospec 100E UV/Visible.

### 5.5. Testing the ASO Toxicity in Human Cell Line

The pVEC-ASO toxicity was tested in a human cell line derived from non-small cell lung cancer A549. The line was cultured as a monolayer culture in a culture medium D-MEM, adding the antibiotics penicillin (100 U/mL), streptomycin (100 µg/mL), and 10% fetal calf serum. The cells from the A549 line were seeded in a 96-well plate (at 100,000 per well), and on the 24th hour from their culturing, in the exponential growth phase, to the culture medium, the ASO was added. The cell survival was established on the 48th hour of the cell treatment with pVEC-ASO by the so-called MTT test, according to Mosmann [28].

## Figures and Tables

**Figure 1 antibiotics-11-01662-f001:**
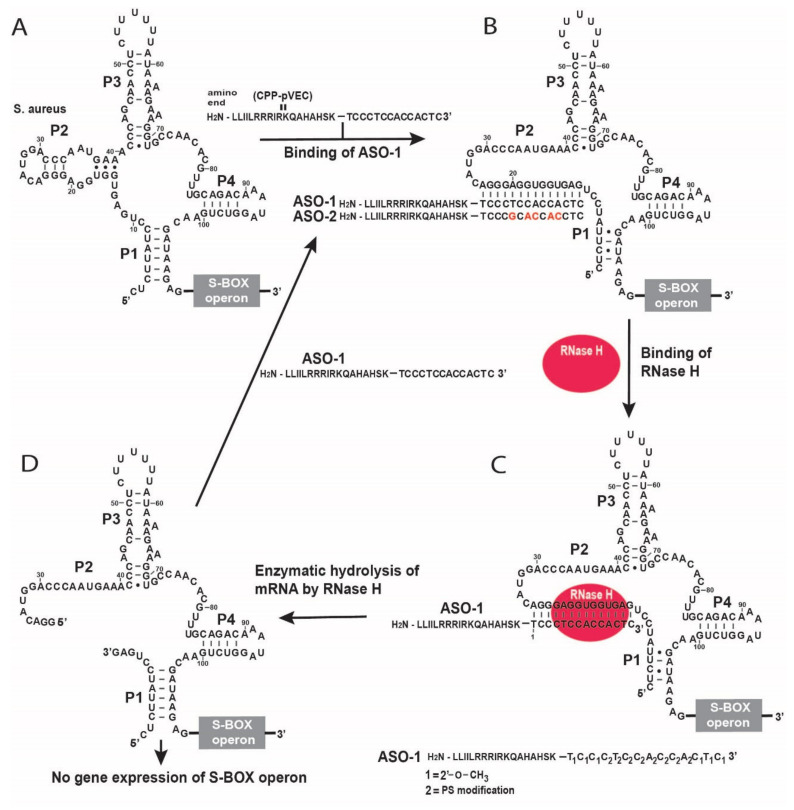
**A chimeric antisense oligonucleotide pVEC-ASO-1 targets the SAM-I aptamer domain of the S-box polycistronic mRNA.** (**A**) The chimeric antisense oligonucleotide binds to the complementary sequence of the SAM-I aptamer domain. (**B**) A double-stranded molecule is formed after binding the chimeric antisense oligonucleotide with the SAM-I aptamer sequence. (**C**) The recognized double-stranded molecule by the RNase H is enzymatically hydrolyzed. (**D**) The enzymatic hydrolysis of RNA leads to no gene expression of the S-box operon.

**Figure 2 antibiotics-11-01662-f002:**
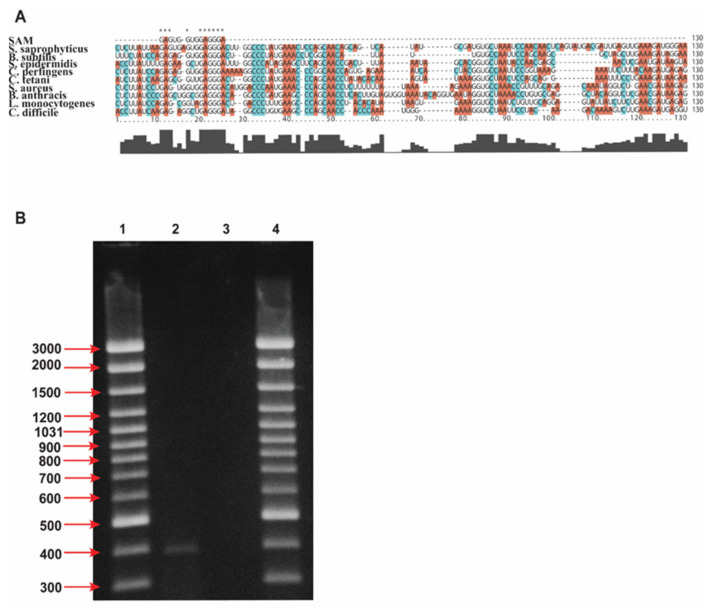
(**A**) **Alignment of the sequences in pathogenic bacteria containing the SAM-I riboswitch sequence.** The pathogenic bacteria *S. aureus, S. sapropyticus, S. epidermidis, L. monocytogenes, C. tetani, C. perfingens, C. difficile, B. anthracis*, and the non-pathogenic bacteria *B. subtilis* contains the SAM-I riboswitch sequence. The same color has the same nucleotide. (**B**) Gel electrophoresis of the SAM-I aptamer RNA from S. aureus. Total RNA is isolated and converted into cDNA. Two primers were used to amplify a SAM-I aptamer region targeted by the pVEC-AS0-1 (1) Determining the size of the DNA by 1000 bp. (2) The amplified cDNA from cells without treatment with pVEC-ASO-1. (3) The amplified cDNA treatment with pVEC-ASO-1.

**Figure 3 antibiotics-11-01662-f003:**
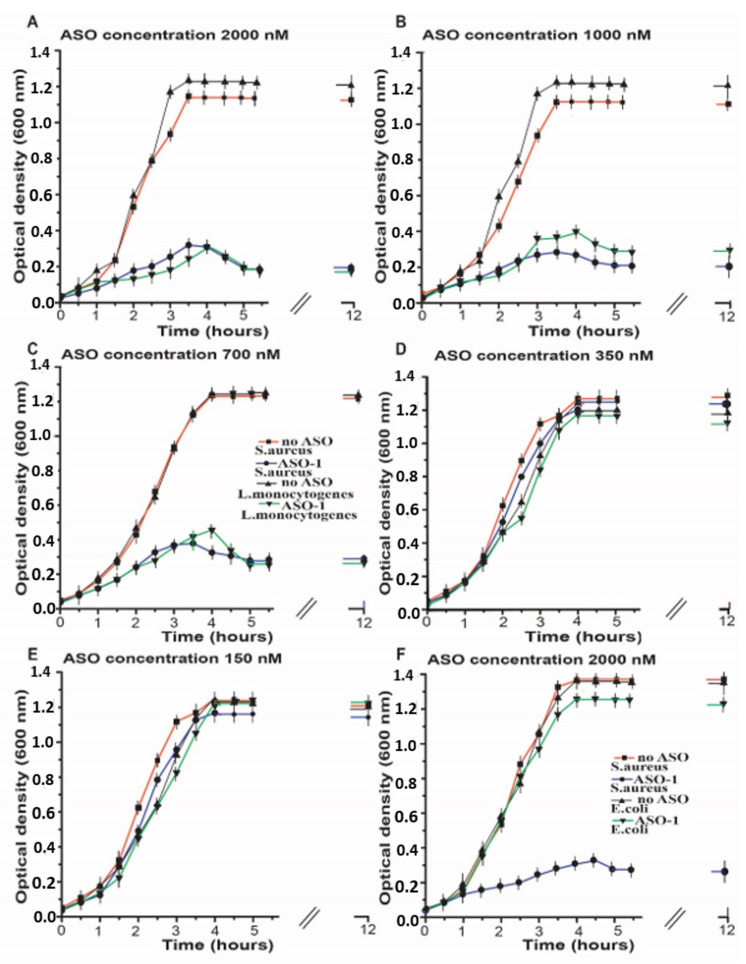
**Inhibition of bacterial growth by antisense oligonucleotide that targets SAM-I riboswitch mRNA.** (**A**) pVEC-ASO-1, 2000 nM, binds with the mRNA for the SAM-I riboswitch and inhibits the bacterial growth of *L. monocytogenes* (the line with the upside-down triangles) and *S. aureus* (the line with the circles). In comparison, without pVEC-ASO-1, there is no inhibition of the bacterial growth of *L. monocytogenes* (the line with the triangles) and *S. aureus* (the line with the rectangles). (**B**) pVEC-ASO-1, 1000 nM, binds with the mRNA for the SAM-I riboswitch and inhibits the bacterial growth of *L. monocytogenes* (the line with the upside-down triangles) and *S. aureus* (the line with the circles). In comparison, without it, there is no inhibition of the bacterial growth of *L. monocytogenes* (the line with the triangles) and *S. aureus* (the line with the rectangles). (**C**) pVEC-ASO-1, 700 nM, also inhibits the bacterial growth of *L. monocytogenes* (the line with the upside-down triangles) and *S. aureus* (the line with the circles). Without pVEC-ASO-1, there is no inhibition of the bacterial growth of *L. monocytogenes* (the line with the triangles) and *S. aureus* (the line with the rectangles). (**D**) At a concentration of 350 nM of pVEC-ASO-1, there is almost no difference in the bacterial growth of *L. monocytogenes* (the line with the upside-down triangles) and *S. aureus* (the line with the circles) as in the presence as well in the absence of ASO-1 (the line with the triangles for *L. monocytogenes*; the line with the rectangles for *S. aureus*). (**E**) At a concentration of 150 nM of pVEC-ASO-1, no difference is observed in the bacterial growth of *L. monocytogenes* (the line with the upside-down triangles) and *S. aureus* (the line with the circles) with and without pVEC-ASO-1 (the line with the triangles for *L. monocytogenes*; the line with the rectangles for *S. aureus*). (**F**) At a concentration of 2000 nM of pVEC-ASO-1, it binds with the mRNA for the SAM-I riboswitch and inhibits the bacterial growth of *S. aureus* (the line with the circles). Without pVEC-ASO-1, there is no inhibition of the bacterial growth of *S. aureus* (the line with the rectangles). Compared to *E. coli*, there is no difference in its bacterial growth in the presence of 2000 nM pVEC-ASO-1 (the line with the upside-down triangles) as in the absence of pVEC-ASO-1(the line with the triangles).

**Figure 4 antibiotics-11-01662-f004:**
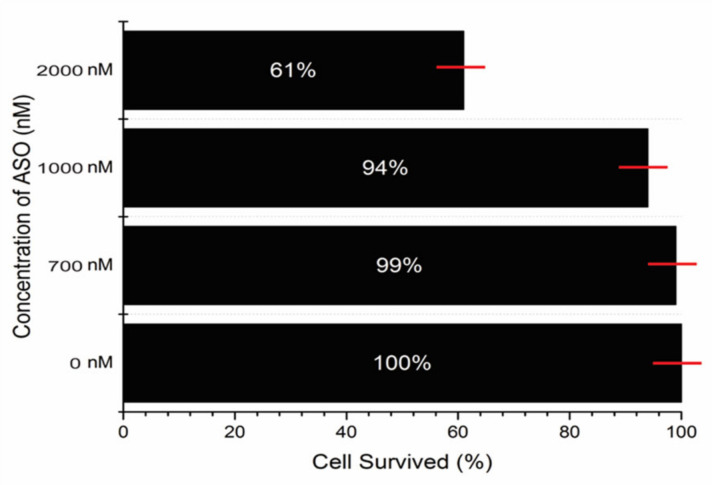
**Probing the pVEC-ASO’s toxicity level in a lung cancer human cell line.** In a concentration of 2000 nM of pVEC-ASO-1, the survival of the cell line is 61%. In the next concentration of 1000 nM, the survival of the cell line is 94%. At a concentration of 700 nM, the survival of the cell line is 99%. In the absence of pVEC-ASO-1, the survival of the cell line is 100%.

**Figure 5 antibiotics-11-01662-f005:**
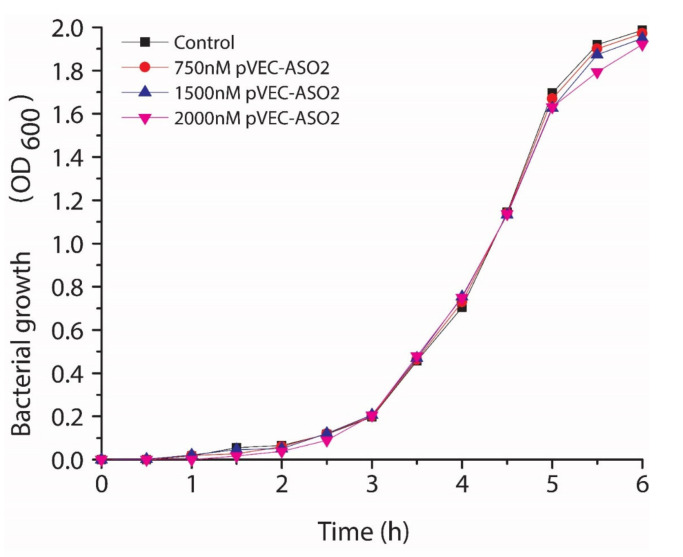
Testing the antibacterial activity of pVEC-ASO-2 with five mismatches compared with pVEC-ASO-1. There was no inhibition of the growth of S. aureus even at 2000 nM.

## Data Availability

https://penchovsky.atwebpages.com/applications.php?page=58 (accessed on 17 November 2022).

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
