# Peer review of "Targeting SAM-I Riboswitch Using Antisense Oligonucleotide Technology for Inhibiting the Growth of Staphylococcus aureus and Listeria monocytogenes"

_antibiotics, 2022, doi:10.3390/antibiotics11111662_

Round 1

Reviewer 1 Report

The manuscript by Traykovska and Penchovsky describes the design of an ASO for targeting SAM-I riboswitch, followed by antibacterial assays. The authors should address the below concerns.

Major comments:

The authors showed the dose dependence of bacterial growth in the presence of the ASO-peptide conjugate. The authors should also show such dose response curves for the mRNA levels (Figure 2B). A control ASO with mutated sequence or scrambled sequence may be added as a control.

Figure 3A and 3B, upside down triangles are found on the top right, which are confusing. The authors may use closed and open symbols for Figure 3.

Line 266, give the detailed calculation method for MIC80.

Line 278, add a non-cancer cell line.

Minor comments:

Line 33, of human lives may be changed to deaths

Line 79, coils may be changed to helixes

Lines 131-134, mention that transcriptional riboswitch regulation is in action.

Figure 2A, show the targeted region by the ASO

Figure 2B caption, add the details of the experimental conditions of the lanes.

Line 198, Figure 3B

Figure 3 caption should be simplified.

Line 271, nucleases should be protease. Give the reference here.

Lines 299-300, add the references.

Line 305, add the reference.

Lines 312-313, peptidase

Line 368, remove reverse

Page S4, Figure 1A should be Figure S1A

Pages S5-S7, Figure S2

Author Response

in the file attached

Reviewer 2 Report

Traykovska and Penchovsky review

The paper describes some preliminary results on the effects of an antisense oligonucleotide,  targeted against the SAM-1 riboswitch mRNA, on the growth of 3 bacteria. Various modifications to the RNA oligonucleotide, as well as its conjugation to an cell penetrating peptide are described here. Inhibition of growth was observed for L. monocytogenes and S. aureus, while E. coli was not inhibited.

I would have liked  if the authors had used a proper control, containing pVEC conjugated to an antisense oligonucleotide that is either scrambled in sequence (with respect to ASO-1), or carries base substitutions predicted to prevent binding to SAM-1.

Figure 3 is needlessly complex:

 A common set of symbols should be used in all panels to indicate growth of S. aureus or L. monocytogenes, with or without pVEC-ASO-1. Additional symbols (diamonds, open circles etc.) can be used for E. coli, or other combinations of bacterium/chemical. Also, the legend seems to cover some of the interpretation also explained in the text. This is needlessly repetitive and should be removed from the legend.

The Discussion section is repetitive. The first paragraph covers material already covered in either the introduction, or first part of the Results. Remove this.

A description of pVEC should appear in the Results section. As it stands, I had to read through to the discussion before I found it described. Move this description (lines 305-315) to line 116 of Results, where it belongs.

The abbreviation CPP (line 125) needs to be described at first mention.

The statement in the Conclusions (line 319) "The pVEC-ASO-1 blocks any alternative pathway of synthesis and transport of  SAM into the cell." is not supported by the data presented here.

I have similar misgivings about another statement from the conclusions (line 326-7) " At the same time, the specific action of pVEC-ASO-1 is proven by the lack of any inhibition in E. coli, where SAM-I riboswitch is not present." There can be many reasons why E. coli did not respond in the same way as S. aureus or L. monocytogenes to the antisense oligo. A proper control oligonucleotide (see above) is needed to support any such claims of specificity.

Line 67: The phrase  'is found' is missing here, after 'riboswitch'.

lines 162-168 are repeated in tandem.

Re-phrase the sentence on lines 184-186 to read  'The maximum inhibition of bacterial growth of S. aureus and L. monocytogenes was observed in the samples containing the highest concentration of pVEC-ASO-1 (2000 nM).

Line 198: remove "and the highest concentration of pVEC-ASO-1".

Author Response

in the file attached

Reviewer 3 Report

In the paper entitled “Targeting SAM-I riboswitch using antisense oligonucleotide technology for inhibiting the growth of Staphylococcus aureus and Listeria monocytogenes” by Martina Traykovska and Robert Penchovsky, the authors designed antisense oligonucleotide against SAM-1 riboswitch and

they found that addition of the antisense oligonucleotide inhibited the growth of the bacteria, L. monocytogenesand S. aureus. However, they observed only growth curves during five hours. They should examine the inhibitory effect during more longer time, because the inhibitory effect is possibly removed after more longer time due to the growth of the survived bacteria. Also, they did not examine whether the designed mechanism really performed in the bacteria to inhibit the target SAM-1 riboswitch. For example, they should measure the mRNA and/or protein level of the gene for the SAM-1 riboswitchand the related metabolism pathway by real-time PCR and/or Western blotting. They should also examine whether the added antisense oligonucleotide inhibited any other genes by off-target effect.

Many scientific mistakes are observed. For the first example, the authors misunderstand that pVEC-ASO-1 binds to the complementary sequence of the aptamer domain of the SAM-1 riboswitch. Because pVEC-ASO-1 itself contains the complementary sequence, it binds to the target sequence of the aptamer domain of the SAM-1 riboswitch (line 1 in the first paragraph of 2.2 of page 4; lines 3 and 9 in the first paragraph of page 8; line 2 from the bottom of the third paragraph of page 9). For the second example, the authors misunderstand that pVEC is degraded by nucleases. Because pVEC is a peptide, it is degraded by peptidases (lines 7 in the first paragraph of page 8; line 3 from the bottom of the third paragraph of page 9). For the third example, “Without pVEC-ASO-1” is incorrect, but “With pVEC-ASO-1” is correct (line 3 of the fourth paragraph of page 6). The authors’ description is not consistent with the result of Figure 3C. This is a careless mistake.

The number of the cited references is small. For example, the authors did not site any papers about chemical modification of nucleic acids, such as phosphorothioate and OCH3 modifications in the result section, and LNA and PNA in the discussion section, although they used phosphorothioate and OCH3 modifications in their experiments. Also, the numbering in the reference list in page 12 is incorrect. References 2, 3, 4, 5, and 6  appear a few times. 

The arrangement in the paper is not appropriate. For the first example, the first paragraph of page 5 is exactly the same as the last paragraph of page 4. For the second example, Figure 1 is observed both in the main text and the supplementary data. For the third example, Figure S2 is missing in the supplementary data. 

Many grammatical errors are found both in main text and supplementary data. The authors should use English editing service before submission. For example, “de” is not English and it should be changed into “be” in line 2 of the second paragraph of page 4. 

On the basis of these considerations about many mistakes, I have to conclude that the present paper is far below the level to justify publication in the journal of Antibiotics

Author Response

in the file attached

Reviewer 4 Report

May be published in Antibiotics after major revision

In their manuscript entitled « Targeting SAM-I riboswitch using antisense oligonucleotide technology for inhibiting the growth of Staphylococcus aureus and Listeria monocytogenes », Martina Traykovska and Robert Penchovsky aim to demonstrate the specific inhibition of the SAM-1 riboswitch by pVEC-ASO-1 for inhibiting the growth of the above-mentioned bacterial pathogens.

General appreciation

In my opinion, this study investigates a promising alternative solution to overcome the loss of activity of antibiotics. Some potentially insightful results are reported in the form of an interesting to read manuscript. However, I also think that some controls are missing in at least one experiment, the presentation of results should be improved and the overall writing revised, especially for referring to previous studies and better delineating the different sections in the manuscript. Accordingly, I recommend a major revision of this manuscript, for taking into account the abovementioned remarks that are further detailed below.

Main comments

From an experimental point of view:

In Section 2.2 “Inhibition of the Bacterial Growth via the S-box mRNA with ASO that targets the SAM-I Riboswitches”, author report the detection via RT-PCR of SAM-1 riboswitch mRNA. It is stated lines 165-168 “Amplification was observed only on the lane with cDNA incubated without ASO-1 (Figure 2B). In contrast, no amplification was observed in the presence of pVec-ASO-1”. This writing may be confusing for the reader. Furthermore, since bacterial growth is expected to be strongly inhibited in the presence of pVec-ASO-1 (according to Figure 3), I wonder how authors managed to collect sufficient amount of RNA and to properly compare with the condition in the absence of pVec-ASO-1. Along this line, in Figure 2 panel B, a positive control for RT-PCR must be included.

In Figure 3, it is unclear why the control consisting of E. coli has not been tested for every dose of pVEC-ASO-1.

A comparison with already reported riboswitches would have been insightful to better appreciate the results reported herein.

As for the writing of the manuscript:

In this paper, only 14 references to previous works have been used. Many statements are not supported with any reference. In several instances, it is unclear whether authors report original or previously reported findings.

In Figure 2 panel A, the meaning of the color code is missing. The sequences of S. aureus and L. monocytogenes should be highlighted and more explanation added.

The writing of Discussion and Conclusion can be improved. Some information should be moved to each other of these two sections. The information about previously reported riboswitches (line 328-329) should be given in Introduction.

In Discussion, Figures should be cited and the results they provide should be discussed in detail in this part.

Other comments

- The reason(s) for using A549 cell line to evaluate cytotoxicity should be given.

- The first paragraph on page 5 looks a duplication of the last paragraph on page 4.

- The caption of most Figures can be improved, especially for providing the relevant information in each case (e.g. by adding the meaning of P1, P2, P3 in Figure 1 caption or on the opposite by removing/shortening the long redundant description in Figure 3 caption).

- Figure 3 should be better designed and the different series better shown (using different colors for example). The legend is confusing.

-  The manuscript must be thoroughly checked to remove typo errors or add missing words. For instances: line 42, “are taken” (“not” is missing); line 136, “to de single-man” (please check); line 284, “cell lines” (delete “s”, since a single cell line was studied).

- Bacterial names are not systematically italicized.

- Line 364, numbers “1” and “2” should be subscripts.

Author Response

in the file attached

Round 2

Reviewer 1 Report

Figure 1C, RNase H should be shown as a transparent cartoon so that the sequences of ASO and riboswitch can be seen.

Figure 3A,B, The authors didn’t get my point. There is a serious mistake that should be fixed. If we look at panels A and B closely, we can see that different symbols (e.g., both triangles and up-side-down triangles within one curve, in another curve both squares and circles within one does-response curve) are found for the same does response curve. The authors should be more careful about data analysis.

Figures S6 caption, without should be with. Give the detailed sequence of ASO2 in Figure 1C (aligned with ASO1). 

Author Response

First

Figure 1C, RNase H should be shown as a transparent cartoon so that the sequences of ASO and riboswitch can be seen.

I do agree. The RNase H in Figure 1C is made a transparent cartoon so that the sequences of ASO and riboswitch can be seen. 

Figure 3A,B, The authors didn’t get my point. There is a serious mistake that should be fixed. If we look at panels A and B closely, we can see that different symbols (e.g., both triangles and up-side-down triangles within one curve, in another curve both squares and circles within one does-response curve) are found for the same does response curve. The authors should be more careful about data analysis.

 I do agree. The symbols between panels A and B are made the same.

Figures S6 caption, without should be with. Give the detailed sequence of ASO2 in Figure 1C (aligned with ASO1). 

I do agree. It is fixed.  

Reviewer 3 Report

The authors removed many scientific and grammatical mistakes. However, they rejected to use English editing service for their present paper. The referee cannot understand the reason. Also, although the referee suggested the following points to improve their present paper, the authors rejected the improvement without showing any clear reasons. They repeated to explain the same results without adding any new experimental results.

<First point>

The authors observed only growth curves during five hours. They should examine the inhibitory effect during more longer time, because the inhibitory effect is possibly removed after more longer time due to the growth of the survived bacteria. Many readers may be interested in how long the inhibitory effect maintain in the bacteria when the readers may use the present antibiotics as drugs. This information is quite important to evaluate the inhibitory effect of the present drug. The authors should make additional experiments for readers’ convenience.

<Second point >

Also, they did not examine whether the designed mechanism really performed in the bacteria to inhibit the target SAM-1 riboswitch. For example, they should measure the mRNA and/or protein level of the gene for the SAM-1 riboswitch and the related metabolism pathway by real-time PCR and/or Western blotting. They should also examine whether the added antisense oligonucleotide inhibited any other genes by off-target effect. Many readers may be interested in whether the designed mechanism really performed in the bacteria and whether the present drug showed any off-target effect or not. The referee understand that the author made one PCR experiment, but it is not enough. Real-time PCR and/or Western blotting experiments are easy. Scientists to develop new drugs usually make these experiments. The information is quite important to evaluate the quality of the present drug. The authors should make additional experiments for readers’ convenience.

The referee would like to judge whether the present paper is suitable for the publication in the journal of “Antibiotics” after the improvement.

Author Response

Third

The authors removed many scientific and grammatical mistakes. However, they rejected to use English editing service for their present paper. The referee cannot understand the reason. Also, although the referee suggested the following points to improve their present paper, the authors rejected the improvement without showing any clear reasons. They repeated to explain the same results without adding any new experimental results.

We used the professional version of Grammarly, which found no problems with the English grammar of the paper.

<First point>

The authors observed only growth curves during five hours. They should examine the inhibitory effect during more longer time, because the inhibitory effect is possibly removed after more longer time due to the growth of the survived bacteria. Many readers may be interested in how long the inhibitory effect maintain in the bacteria when the readers may use the present antibiotics as drugs. This information is quite important to evaluate the inhibitory effect of the present drug. The authors should make additional experiments for readers’ convenience.

We have shown the curves for 12 hours – see Figure 3.

<Second point >

Also, they did not examine whether the designed mechanism really performed in the bacteria to inhibit the target SAM-1 riboswitch. For example, they should measure the mRNA and/or protein level of the gene for the SAM-1 riboswitch and the related metabolism pathway by real-time PCR and/or Western blotting. They should also examine whether the added antisense oligonucleotide inhibited any other genes by off-target effect. Many readers may be interested in whether the designed mechanism really performed in the bacteria and whether the present drug showed any off-target effect or not. The referee understand that the author made one PCR experiment, but it is not enough.

 Real-time PCR and/or Western blotting experiments are easy. Scientists to develop new drugs usually make these experiments. The information is quite important to evaluate the quality of the present drug. The authors should make additional experiments for readers’ convenience.

Our PCR results show a specific amplification of SAM-I mRNA  without a treatment with pVEC-ASO-1 and no amplification after treatment with pVEC-ASO-1 Fig. 2B.  The real-time PCR does not add anything more than that. Moreover, it cannot show the specific amplification without the treatment with pVEC-ASO-1, and it can show just amplification.

We disagree with the necessity to show real-time PCR data because they do not add anything to the existing data. We cannot detect the presence of SAM-I mRNA when the cells are treated with 2000 nM pVEC-ASO-1 with any PCR, including real-time PCR with SYBRGreen I.

Reviewer 4 Report

In their revised manuscript entitled « Targeting SAM-I riboswitch using antisense oligonucleotide technology for inhibiting the growth of Staphylococcus aureus and Listeria monocytogenes », Martina Traykovska and Robert Penchovsky took into account some, but not all, the comments provided. Control conditions are critical to convince the reader about the results reported, thus corresponding data would better fit in the main manuscript than in supplementary information. Beside this, the paper is still poorly written with hard to read Figures; it is regrettable that Authors did not took advantages of the benevolent comments received (from myself as well as the other Reviewers) to really improve their manuscript; careless mistakes are still present. Accordingly, I have to recommend again a major revision of this manuscript, for addressing all the points that are noted again below as well as some additional ones.

---

Since bacterial growth is expected to be strongly inhibited in the presence of pVec-ASO-1 (according to Figure 3), I wonder how authors managed to collect sufficient amount of RNA and to properly compare with the condition in the absence of pVec-ASO-1. PLEASE ADDRESS THIS POINT.

In Figure 2 panel B, a positive control for RT-PCR must be included. PLEASE ADDRESS THIS POINT.

A comparison with already reported riboswitches would have been insightful to better appreciate the results reported herein. PLEASE ADDRESS THIS POINT.

Figure 3 should be better designed and the different series better shown (using different colors for example). The legend is still confusing. PLEASE ADDRESS THIS POINT, USING COLORS (AS DONE IN FIGURE S4-S6) OR REFORMATTING THE FIGURE, WITH CLEARLY IDENTIFIABLE SERIES SO IT IS READABLE AND ACTUALLY USEFUL.

In Discussion, Figures should be cited and the results they provide should be discussed in detail, in an objective and impartial manner (without overstating the results, and avoiding sentences such as "we believe ..." or "it is improbable that..."). PLEASE ADDRESS THIS POINT.

FIGURE S6 LACKS EXPLANATION, MISMATCHES COMPARED WITH ASO-1 ARE NOT HIGHLIGHTED AND IMPORTANT EXPLANATIONS ARE HERE ALSO LACKING. THE CAPTION IS INCORRECT.

Author Response

Fourth

In their revised manuscript entitled « Targeting SAM-I riboswitch using antisense oligonucleotide technology for inhibiting the growth of Staphylococcus aureus and Listeria monocytogenes », Martina Traykovska and Robert Penchovsky took into account some, but not all, the comments provided. Control conditions are critical to convince the reader about the results reported, thus corresponding data would better fit in the main manuscript than in supplementary information. Beside this, the paper is still poorly written with hard to read Figures; it is regrettable that Authors did not took advantages of the benevolent comments received (from myself as well as the other Reviewers) to really improve their manuscript; careless mistakes are still present. Accordingly, I have to recommend again a major revision of this manuscript, for addressing all the points that are noted again below as well as some additional ones.

We used the professional version of Grammarly, which found no problems with the English grammar of the paper.

The figure S6 from SI is moved to the main body of the paper as figure 5.

---

Since bacterial growth is expected to be strongly inhibited in the presence of pVec-ASO-1 (according to Figure 3), I wonder how authors managed to collect sufficient amount of RNA and to properly compare with the condition in the absence of pVec-ASO-1. PLEASE ADDRESS THIS POINT.

We added in the Experimental section:”Total RNAs were isolated from the same amount of cells with 0.15 OD treated with or without pVEC-ASO-1.”  

In Figure 2 panel B, a positive control for RT-PCR must be included. PLEASE ADDRESS THIS POINT.

The positive control is already included – see Figure 2B, lane 2. It means amplification of a region of SAM-I mRNA when the cell are not treated with pVEC-ASO-1.

A comparison with already reported riboswitches would have been insightful to better appreciate the results reported herein. PLEASE ADDRESS THIS POINT.

We added in Conclusion : “Designer pVEC-ASOs targeting FMN[9], TPP[10], and SAM-I riboswitches have MIC80 around 700nM and exhibit a bacteriostatic effect. All designer pVEC-ASOs worked as expected, proving the high fidelity of our rational approach for drug design, including drug-target evaluation [6,7]. FMN, TPP, and SAM-I are essential co-factors for many enzymes in the cell, and it takes 3 to 5 hours after stopping their syntheses till the inhibition of bacterial growth. The bacteria may use already synthesized FMN, TPP, or SAM-I within this period of 3 to 5 hours. In contrast, the complete inhibition of glmS mRNA by pVEC-ASOs has a bacteriocidal effect due to the inhibition of cellualr wall synthesis [8]. By targeting riboswitches with ASOs, we can develop broad[9] and nar-row-range[8] antibiotics.”

We will discussed that in more detail in a full review paper soon.

Figure 3 should be better designed and the different series better shown (using different colors for example). The legend is still confusing. PLEASE ADDRESS THIS POINT, USING COLORS (AS DONE IN FIGURE S4-S6) OR REFORMATTING THE FIGURE, WITH CLEARLY IDENTIFIABLE SERIES SO IT IS READABLE AND ACTUALLY USEFUL.

I do agree. In Figure 3, we use different colors.

In Discussion, Figures should be cited and the results they provide should be discussed in detail, in an objective and impartial manner (without overstating the results and avoiding sentences such as "we believe ..." or "it is improbable that..."). PLEASE ADDRESS THIS POINT.

We have removed this from the discussion “As a result, all designed by us pVEC-ASOs worked as planned, making 100% efficiency out of 4 targets. We believe this consistent success is not random but based on our general rational strategy for designing chimeric pVEC-ASOs that target bioinformatically chosen riboswitches in various pathogenic bacteria.”

FIGURE S6 LACKS EXPLANATION, MISMATCHES COMPARED WITH ASO-1 ARE NOT HIGHLIGHTED AND IMPORTANT EXPLANATIONS ARE HERE ALSO LACKING. THE CAPTION IS INCORRECT.

It is corrected to ”Testing the antibacterial activity of pVEC-ASO-2 with 5 mismatches compared with pVEC-ASO-1. There is no inhibition of the growth of S. aureus even at 2000 nM.

Round 3

Reviewer 4 Report

The manuscript by Martina Traykovska and Robert Penchovsky has been substantially revised by taking into account many of the recommendations provided.

However:

- A positive control for RT-PCR is still lacking. An additional target RNA that is expressed in every sample assayed shoud have been studied in parrallel, thus demonstrating that the absence of amplification in lane 3 is not due to any experimental issue.

- As for the writing of the manuscript and the quality of presentation, I think there is still ample room for improvements.

At this stage, I leave the authors consider the utility of the comments provided that imo will determine the impact of their study.

Author Response

- A positive control for RT-PCR is still lacking. An additional target RNA that is expressed in every sample assayed shoud have been studied in parrallel, thus demonstrating that the absence of amplification in lane 3 is not due to any experimental issue.

  1. I have to write more to make. Both cDNA and PCR experiments with and without pVEC-ASO-1 are done with the same master mix, which is divided into two equal portions. That is a standard pipetting practice in our lab to reduce the pipetting needed for one set of experiments. That excluded all errors in the PCR due to different conditions between lanes 2 and 3 in Figure 2B. That is now written in the text in ES and in the results. In addition, cDNAs  are tested with primers for FMN mRNA in Figure S6. That addressed all possible positive controls.  

- As for the writing of the manuscript and the quality of presentation, I think there is still ample room for improvements.

We have made a lot of edits given as yellow highlight.